**Data Availability Statement:** All relevant data are within the manuscript and its Supporting Information files.

**Funding:** This research was supported by a National Research Foundation of Korea (NRF) grant

# Enhanced cytotoxic activity of natural killer cells from increased calcium influx induced by electrical stimulation

**Minseon Lee**[1,2], **Soonjo Kwon**[1,2]*

1 Department of Biological Engineering, Inha University, Incheon, Korea, 2 Department of Biological Sciences and Bioengineering, Inha University, Incheon, Korea

* soonjo.kwon@inha.ac.kr

## Abstract

Natural killer (NK) cells play a crucial role in immunosurveillance independent of antigen presentation, which is regulated by signal balance via activating and inhibitory receptors. The anti-tumor activity of NK cells is largely dependent on signaling from target recognition to cytolytic degranulation; however, the underlying mechanism remains unclear, and NK cell cytotoxicity is readily impaired by tumor cells. Understanding the activation mechanism is necessary to overcome the immune evasion mechanism, which remains an obstacle in immunotherapy. Because calcium ions are important activators of NK cells, we hypothesized that electrical stimulation could induce changes in intracellular $Ca^{2+}$ levels, thereby improving the functional potential of NK cells. In this study, we designed an electrical stimulation system and observed a correlation between elevated $Ca^{2+}$ flux induced by electrical stimulation and NK cell activation. Breast cancer MCF-7 cells co-cultured with electrically stimulated KHYG-1 cells showed a 1.27-fold (0.5 V/cm) and 1.55-fold (1.0 V/cm) higher cytotoxicity, respectively. Electrically stimulated KHYG-1 cells exhibited a minor increase in $Ca^{2+}$ level (1.31-fold (0.5 V/cm) and 1.11-fold (1.0 V/cm) higher), which also led to increased gene expression of granzyme B (GZMB) by 1.36-fold (0.5 V/cm) and 1.58-fold (1.0 V/cm) by activating $Ca^{2+}$-dependent nuclear factor of activated T cell 1 (NFAT1). In addition, chelating $Ca^{2+}$ influx with 5 μM BAPTA-AM suppressed the gene expression of $Ca^{2+}$ signaling and lytic granule (granzyme B) proteins by neutralizing the effects of electrical stimulation. This study suggests a promising immunotherapeutic approach without genetic modifications and elucidates the correlation between cytolytic effector function and intracellular $Ca^{2+}$ levels in electrically stimulated NK cells.

## Introduction

Immunotherapy, such as adoptive cell therapy (ACT), has emerged as one of the most promising cancer treatments, with even higher potential than existing traditional chemotherapy. The discovery of immune checkpoints (ICs), such as CTLA-4, PD-1, or the recently identified B7-H3 and its inhibitors, offers an approach for targeting new therapeutic candidates or

funded by the Korea government (MSIT) (RS-2023-00207801) and an Inha University Research Grant, Korea. The funders had no role in study design, data collection and analysis, decision to publish, or preparation of the manuscript.

**Competing interests:** The authors have declared that no competing interests exist.

efficiently combining immunotherapies with increasing dimensions [1–3]. In contrast to T cell therapy, which carries the risk of cytokine storm syndrome (CRS) or graft-versus-host disease (GVHD), the inherent properties of natural killer (NK) cells enable cancer immunotherapy with higher stability and off-the-shelf utility with unlimited allogeneic NK sources [4–6]. However, owing to the challenges of *in vivo* persistence, poor infiltration into solid tumors, and immunosuppressive environments such as hypoxia or various soluble factors in tumor cells, diverse engineering methods are being developed to overcome the limitations of NK cells for further therapeutic applications [7–9].

NK cells are crucial for innate immunity and exhibit a wide range of major functions, including cytotoxic activity, cytokine production, and homing capabilities [10]. Activated NK cells in the "missing-self recognition" response induce tumor cell death via two major pathways, the release of perforin/granzyme-containing granules and/or death receptor-mediated apoptosis by tumor necrosis factor (TNF) family ligands [11]. The early stages of killing are predominately mediated by granzyme B in cytolytic granules in the serial killing mechanism of NK cells, followed by caspase-mediated apoptosis [12]. In this process, an increase in dense-core secretory granules with either more degranulation or higher granule content was recently reported to be associated with improved NK cell effector functions [13]. $Ca^{2+}$ influx through calcium release-activated calcium (CRAC) channels is crucial for tuning killer cell function [14–16]. $Ca^{2+}$-sensitive nuclear factor of activated T cells (NFAT) appears to be an essential transcriptional factor that regulates NK cell maturation and anti-tumor functions [17]. We initially hypothesized that targeted manipulation of intracellular $Ca^{2+}$ levels by electrical stimulation could also affect the cytolytic activity of NK cells by modulating cell-intrinsic factors such as transcriptional factors. In previous studies, electrical stimulation has been used to regulate cellular functions such as proliferation, migration, and differentiation of mesenchymal stem cells (MSCs) by modulating $Ca^{2+}$ entry and activating calcineurin-NFAT signaling [18, 19].

In this study, we designed an electrical stimulation system to improve the functional phenotype of NK cells by inducing changes in $Ca^{2+}$ flux. Subsequent analyses showed that marginally enhanced $Ca^{2+}$ levels triggered by electrical stimulation increased lytic granule-mediated effector function via the upregulation of granzyme B expression. The results of this study support the association between intracellular $Ca^{2+}$ and NK cell cytotoxicity, as previously described. These findings highlight the possibility of using electrical simulation as a method for engineering immunotherapy in an intracellular signal-dependent manner without genetic modification.

## Materials and methods

### Cell culture

Human breast cancer cells MCF-7 and MDA-MB-231 obtained from the Korean Cell Line Bank (KCLB; Seoul, Korea) were cultured at 37˚C with 5% $CO_2$ in RPMI 1640 (Gibco, Carlsbad, CA, USA) supplemented with 10% fetal bovine serum (FBS; Gibco) and 1% penicillin-streptomycin (Gibco). The human NK cell line, KHYG-1, was obtained from AcceGen Biotech (cat. # ABC-TC0506; Fairfield, NJ, USA). KHYG-1 cells were cultured in RPMI 1640 medium supplemented with heat-inactivated FBS, 1% penicillin-streptomycin, and 100 units/mL recombinant human IL-2 (cat. # 200–02; PeproTech, Rocky Hill, NJ, USA). KHYG-1 cells were seeded in T-25 flasks and added to fresh medium or subcultured every 2–3 days.

### Exposure to electrical stimulation

The electrical stimulation system was designed and adapted to a cell culture plate, and its performance was verified using an oscilloscope (DSOX1102A; Keysight, Santa Rosa, CA, USA).

The electrical system chamber was printed using a 3D printer (3DP-110F; Cubicon, Seong-nam, Korea) fitted with a lid for standard 6-well plates (Fig 1A). The electrical stimulation system consisted of two platinum (Pt) electrodes (cathodes and anodes) (99.9% Pt wire with 0.3 mm diameter) at a distance of 20 mm in parallel (Fig 1). The condition of stimulation was controlled by the function generator. It was designed to affect cells directly through the culture medium by inserting electrodes, which were fully applied throughout the well. In the electrical stimulation experiments, NK cells were exposed to two voltage ranges (0.5 V/cm, 1.0 V/cm) of direct current (referred to as DC) electrical stimulation or biphasic electrical stimulation for 1 h at a frequency of 10 Hz (symmetrical, charge-balanced, referred to as BP). After the tumor cells reached confluence, they were co-cultured for 24 h with NK cells that had been electrically stimulated for 1 h in advance and compared with the control group without electrical stimulation (Fig 1B). Stimulation with DC conditions was applied for all experiments except the cell viability assay (CCK-8, Fig 2A). All subsequent mentions of the abbreviation 'ES' indicate electrical stimulation with DC conditions. In some experimental sets for chelating $Ca^{2+}$, KHYG-1 cells were incubated 30 min before and during electrical stimulation (1 h) in a medium supplemented with DMSO containing the cell-permeant $Ca^{2+}$ chelator BAPTA-AM (B6769; Invitrogen, Carlsbad, CA, USA) at a final concentration of 5 μM or an equivalent volume of DMSO as the control group.

## Cell viability

The effect of electrical stimulation on cell viability was measured using a cell counting kit-8 (CCK-8) assay (Dojindo Laboratories, Kumamoto, Japan). KHYG-1 cells exposed to electrical stimulation for 1 h were transferred to 96-well plates (100 μL, 150,000 cells) and incubated with 10 μL of CCK-8 at 37°C for 3 h. The absorbance of each well was measured at 450 nm using a microplate reader (Thermo Fisher Scientific).

## Cytotoxicity assay

For the lactate dehydrogenase (LDH) cytotoxicity assay, the medium was collected from each well after KHYG-1 cell treatment at an effector-to-target (E:T) ratio of 3:1 (24 h) and centrifuged at 600 $g$ for 5 min. NK cell-mediated cytotoxicity was measured according to previously described protocols [20] and calculated according to the manufacturer's instructions (Fig 1B).

A live/dead cytotoxicity assay (L3224; Invitrogen) was used to measure the NK cell cytotoxicity at 6 h timepoint during co-culture, because most dead MCF-7 cells can be detached 24 hours after co-culture. The target MCF-7 cells were cultured in 12-well plates and co-cultured with stimulated or non-stimulated effector cells at an E:T ratio of 3:1 for 6 h. MCF-7 cells were washed three times with DPBS to remove KHYG-1 cells, followed by staining with 2 μM calcein AM (live) and 4 μM ethidium homodimer-1 (dead) and analyzed by fluorescence microscopy according to the manufacturer's protocol.

## Gene expression analysis

Reverse transcription-quantitative polymerase chain reaction (RT-qPCR) was used to analyze the effects of electrical stimulation on the expression of cytotoxic granules, cytokine proteins, and $Ca^{2+}$-calcineurin-NFAT signaling proteins. Total RNA from the control and electrically stimulated cell samples was extracted using TRIzol reagent (Life Technologies, Carlsbad, CA, USA), following the manufacturer's recommended protocol. The mRNA concentration of each sample was determined using a microspectrophotometer (DS-11; DeNovix, Wilmington, DE, USA), and cDNA was synthesized using a PrimeScript RT reagent kit (Takara, Shiga, Japan). TB Green Premix Ex Taq II (RR810A; Takara) was applied using a CFX96 detection system

**(A)**

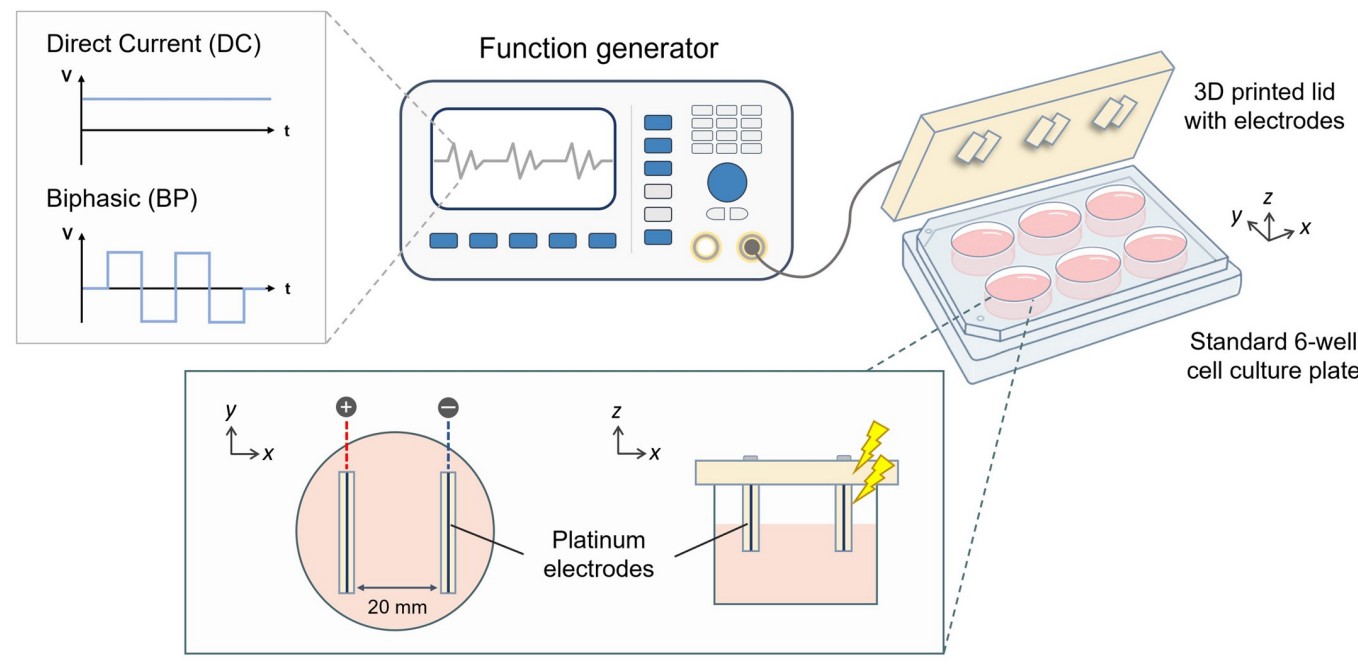

**(B)**

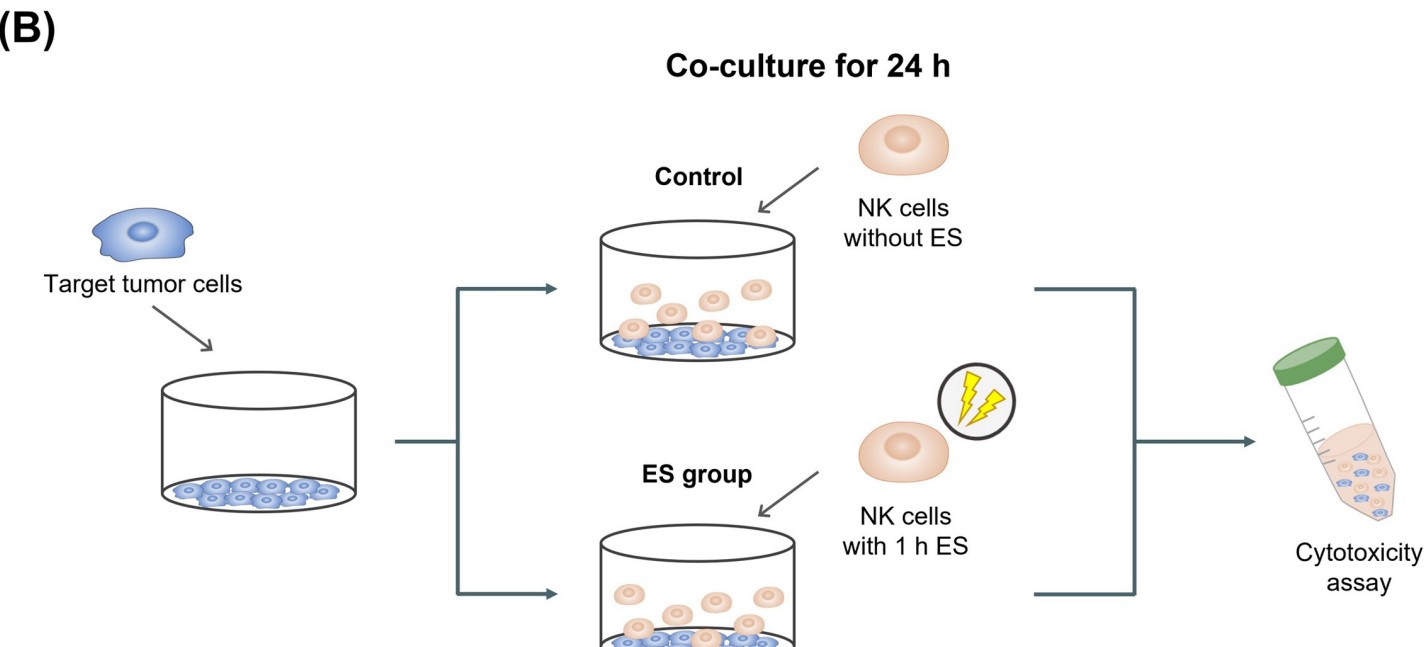

**Fig 1. Electrical stimulation device and experimental setup used in this study.** (A) Schematic representation of the electrical stimulation device with the chamber and two incorporated electrodes. (B) Experimental design of co-culture with NK cells and target tumor cells to analyze the effect of electrical stimulation on the cytolytic activity of NK cells.

(Bio-Rad) for real-time analysis. The target genes were perforin (PRF1), granzyme B (GZMB), interferon-gamma (IFNG), and tumor necrosis factor alpha (TNFA), which are related to the

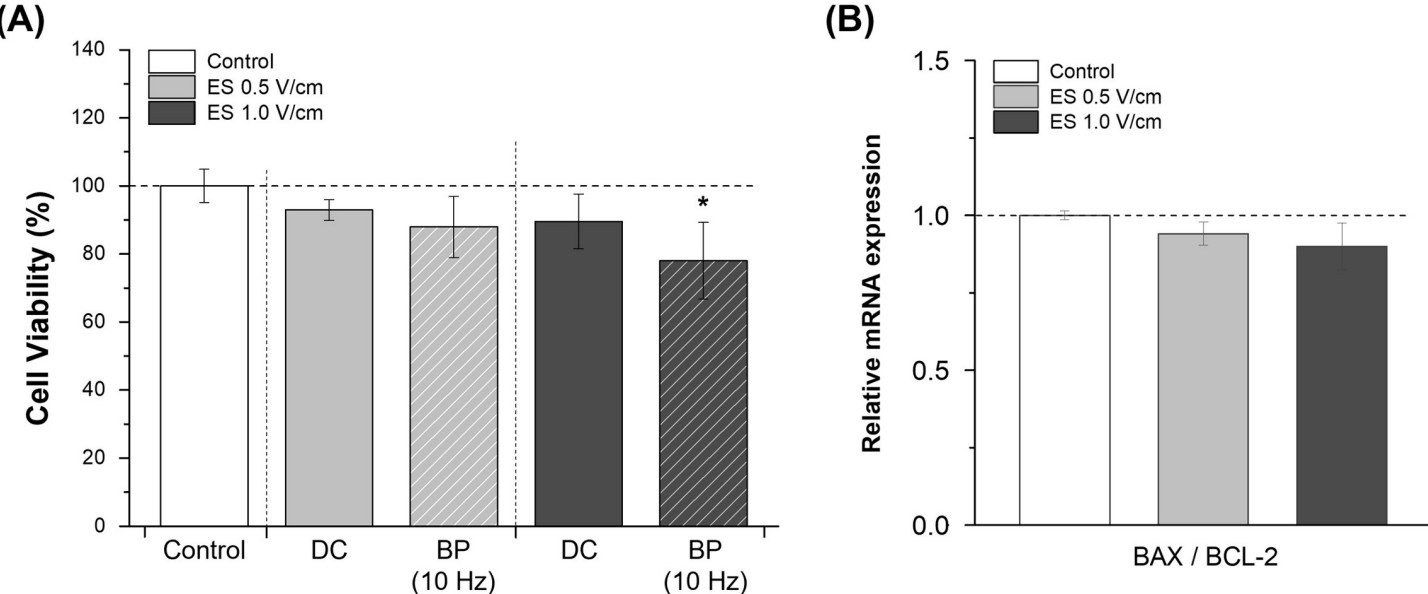

**Fig 2. Effects of electrical stimulation on NK cell viability.** (A) CCK-8 cell viability assays were conducted after exposure to electrical stimulation with constant direct current (DC) or a biphasic (BP) square waveform (10 Hz frequency, column with slashed line) for 1 h (n = 3). (B) The relative mRNA expression ratio of the apoptosis-related genes BAX/BCL-2 was evaluated using RT-qPCR after 1 h of constant DC electrical stimulation (n = 3). Results are presented with average and error bars. Student's t-test was used to determine the statistical significance of the results (*, $P < 0.05$; **, $P < 0.01$ and ***, $P < 0.001$).

effector function of NK cells. Other factors related to calcium signaling include inositol trisphosphate receptor (IP3R), calmodulin (CALM2), calcineurin (CALN), calmodulin-dependent protein kinase II (CaMK II), and nuclear factor of activated T cells (NFAT), which are involved in the $Ca^{2+}$ signaling pathway. GAPDH was used as the housekeeping gene, and the relative mRNA expression level compared to the static control group was calculated using the $2^{(-\Delta\Delta CT)}$ method. The primer sequences for the target genes are listed in Table 1.

### Enzyme-linked immunosorbent assay (ELISA)

KHYG-1 cells were seeded in 6-well plates and incubated at 37°C with 5% $CO_2$ for 4 h and 8 h after stimulation for 1 h under each electrical condition (0.5 V/cm, 1.0 V/cm). The culture medium was collected and centrifuged at 600 $g$ for 5 min to obtain supernatant samples. Intracellular proteins were extracted using radioimmunoprecipitation assay (RIPA) lysis buffer (Elpis Biotech, Daejeon, Korea) containing a protease inhibitor cocktail (Thermo Fisher Scientific) according to the manufacturer's protocol. The granzyme B protein level intracellularly or secreted into the medium was detected using a Human Granzyme B ELISA kit (3486-1H-6; MABTECH, Nacka Strand, Sweden), following the manufacturer's protocol.

### Fluo-4 calcium fluorescence assay

Changes in intracellular $Ca^{2+}$ levels were quantitatively detected using the Fluo-4 NW Calcium Assay Kit (F36206; Invitrogen). KHYG-1 cells were incubated with Fluo-4 AM dye for 45 min at 37°C while being exposed to each voltage range of electrical stimulation, which was performed during the maximum time recommended in the manufacturer's protocol for staining dye. A Varioskan LUX multimode microplate reader (Thermo Fisher) was used to measure the fluorescence intensity (494 nm excitation; 516 nm emission) after transferring 100 μL to black-wall, clear-bottom 96-well plates following the manufacturer's protocol. Data were

**Table 1. Primer sequences of target genes.**

| Target gene | Size (bp [a]) | Sequences | Tm [b] (°C) |
|---|---|---|---|
| GAPDH | 112 | F: CTCCTGTTCGACAGTCAGCC<br>R: CGCCCAATACGACCAAATCC | 60.39<br>59.34 |
| BCL2 | 131 | F: CAACATCGCCCTGTGGATGAC<br>R: GCCAGGAGAAATCAAACAGAGGC | 61.61<br>61.98 |
| BAX | 103 | F: TCAGGATGCGTCCACCAAGAAG<br>R: TGTGTCCACGGCGGCAATCATC | 62.57<br>65.91 |
| PRF1 | 133 | F: ACTCACAGGCAGCCAACTTTGC<br>R: CTCTTGAAGTCAGGGTGCAGCG | 64.10<br>63.66 |
| GZMB | 122 | F: CGACAGTACCATTGAGTTGTGCG<br>R: TTCGTCCATAGGAGACAATGCCC | 62.23<br>62.32 |
| IFNG | 124 | F: GAGTGTGGAGACCATCAAGGAAG<br>R: TGCTTTGCGTTGGACATTCAAGTC | 60.62<br>62.79 |
| TNFA | 108 | F: GCTGCACTTTGGAGTGATCG<br>R: TCACTCGGGGTTCGAGAAGA | 59.55<br>60.25 |
| IP3R | 143 | F: GTGACAGGAAACATGCAGACTCG<br>R: CAGCAGTTGCACAAAGACAGGC | 61.95<br>63.02 |
| CALM2 | 155 | F: AGTGCTGCAGAACTTCGCCATG<br>R: CAAGGTCTTCACTTTGCTGTCATC | 63.94<br>60.32 |
| CALN | 104 | F: GCCCTGATGAACCAACAGTTCC<br>R: GCAGGTGGTTCTTTGAATCGGTC | 61.72<br>62.21 |
| CAMK II | 108 | F: GAGCCATTCTCACCACGATGCT<br>R: TGGTGTTGGTGCTCTCTGAGGA | 62.94<br>63.28 |
| NFAT1 | 125 | F: GATAGTGGGCAACACCAAAGTCC<br>R: TCTCGCCTTTCCGCAGCTCAAT | 61.68<br>64.82 |
| NFAT2 | 132 | F: CACCAAAGTCCTGGAGATCCCA<br>R: TTCTTCCTCCCGATGTCCGTCT | 61.69<br>62.87 |
| NFAT4 | 122 | F: AGACAGTCGCTACTGCAAGCCA<br>R: GCGGAGTTTCAAAATACCTGCAC | 64.19<br>60.91 |

[a] Base pair

[b] Melting temperature (°C)

calculated as relative fluorescence over the baseline and expressed as fold changes in percentage relative to the untreated control. Baseline fluorescence was repeatedly established at the beginning and end of the measurement, according to the manufacturer's protocol. In addition, KHYG-1 cells stained with Fluo-4 AM were independently visualized by fluorescence microscopy and quantitative assays.

## Western blotting

NFAT1 dephosphorylation was analyzed by western blot following exposure to electrical stimulation (1.0 V/cm) or electrical stimulation with BAPTA-AM. After stimulation, the KHYG-1 cells were washed twice with cold DPBS and lysed with RIPA buffer (Elpis Biotech) containing a protease and phosphatase inhibitor cocktail (Thermo Fisher Scientific), followed by incubation on ice for 15 min and sonication for 30 s. Total cell lysates were centrifuged for 15 min and transferred for further analysis. Each sample (10 μL) was diluted in the Laemmli sample buffer and boiled at 95°C for 5 min. After resolution by electrophoresis on a 10% sodium dodecyl sulfate-polyacrylamide gel, the protein samples were transferred to a polyvinylidene difluoride (PVDF) membrane (162–0177; Bio-Rad) using a semi-dry transfer method (Bio-Rad). The membrane was blocked with 5% non-fat dry milk in Tris-buffered saline (TBS) for 1 h at room temperature (25°C) to prevent non-specific binding. The membrane was incubated

overnight at 4˚C with primary antibodies against NFAT1 (MA1-025; Invitrogen) and GAPDH (2118S; Cell Signaling Technology, Danvers, MA, USA) as loading controls. The membranes were probed with horseradish peroxidase-conjugated secondary antibodies (anti-rabbit, ab6721, Abcam; anti-mouse, 31430, Thermo Fisher). Images were acquired using a chemiluminescence substrate (W3651-012; GenDEPOT, Barker, TX, USA) and quantified using a chemiluminescence imaging system (G:BOX Chemi XRQ; Syngene, Cambridge, UK) and Syngene GeneTools software. Dephosphorylated NFAT1 levels were calculated as dephospho-NFAT1 versus phosphorylated NFAT1 (p-NFAT1) and compared with the control group.

## Statistical analysis

All experiments were independently repeated at least three times, and the results are expressed as the mean ± standard error of the mean. Statistical analyses were performed using Student's t-test in Microsoft Excel 2016 (two-tailed, equal variance) to determine the statistical significance of the results obtained between each test group and the control group (* $P < 0.05$, ** $P < 0.01$, and *** $P < 0.001$).

## Results

### Effects of electrical stimulation on cell viability of NK cells

The effect of electrical stimulation on cell viability was measured using CCK-8 assays after 1 h of electrical stimulation. Cell viability was marginally reduced after continuous electrical stimulation with DC, but the difference was not statistically significant (Fig 2A). The long-term effects of electrical stimulation on cell viability and proliferation were also detected using the CCK-8 assay after 24 h and 48 h of stimulation (S1 Fig), which showed no significant differences compared to the control group. Consistent with this result, the quantity of LDH released did not increase in cells electrically stimulated with DC after subsequent incubation for 24 h (S1 Fig). The cells exposed to electrical stimulation with a biphasic square waveform at a frequency of 10 Hz exhibited decreased cell viability, which was significantly lower under electrical stimulation at 1.0 V/cm (Fig 2A). Therefore, subsequent experiments were conducted under continuous electrical stimulation with DC for 1 h. Furthermore, the expression levels of apoptosis-related genes, such as BAX (pro-apoptotic) and BCL2 (anti-apoptotic), were detected using RT-qPCR. The BAX/BCL2 ratio slightly decreased after electrical stimulation with continuous DC, but the decrease was not statistically significant (Fig 2B). Thus, these results indicated that biphasic electrical stimulation significantly decreased cell viability, whereas DC stimulation did not. Subsequent experiments were conducted using continuous electrical stimulation at two different voltages under DC conditions for 1 h.

### Effects of electrical stimulation on the cytotoxic activity of NK cells

To further examine the effect of electrical stimulation on the killing activity of effector NK cells (E) against target tumor cells (T), cell-mediated cytotoxicity was evaluated using an LDH cytotoxicity assay. MDA-MB-231 and MCF-7 cells were cultured in 12-well plates and co-cultured with KHYG-1 cells at an E:T ratio of 3:1 for 24 h (Fig 1B). MDA-MB-231 and MCF-7 cells were co-cultured with KHYG-1 cells pretreated for 1 h in the presence or absence of electrical stimulation at 0.5 V/cm or 1.0 V/cm voltages. The MDA-MB-231 cells with electrically stimulated KHYG-1 showed no significant increase in cytotoxicity (Fig 3A). However, the cytotoxicity of MCF-7 with electrically stimulated KHYG-1 cells increased from 53% to 74% (1.27-fold, 0.5 V/cm electrical stimulation) and from 58% to 90% (1.55-fold, 1.0 V/cm electrical stimulation), respectively. To assess NK cell cytotoxicity, live/dead assay imaging of MCF-7

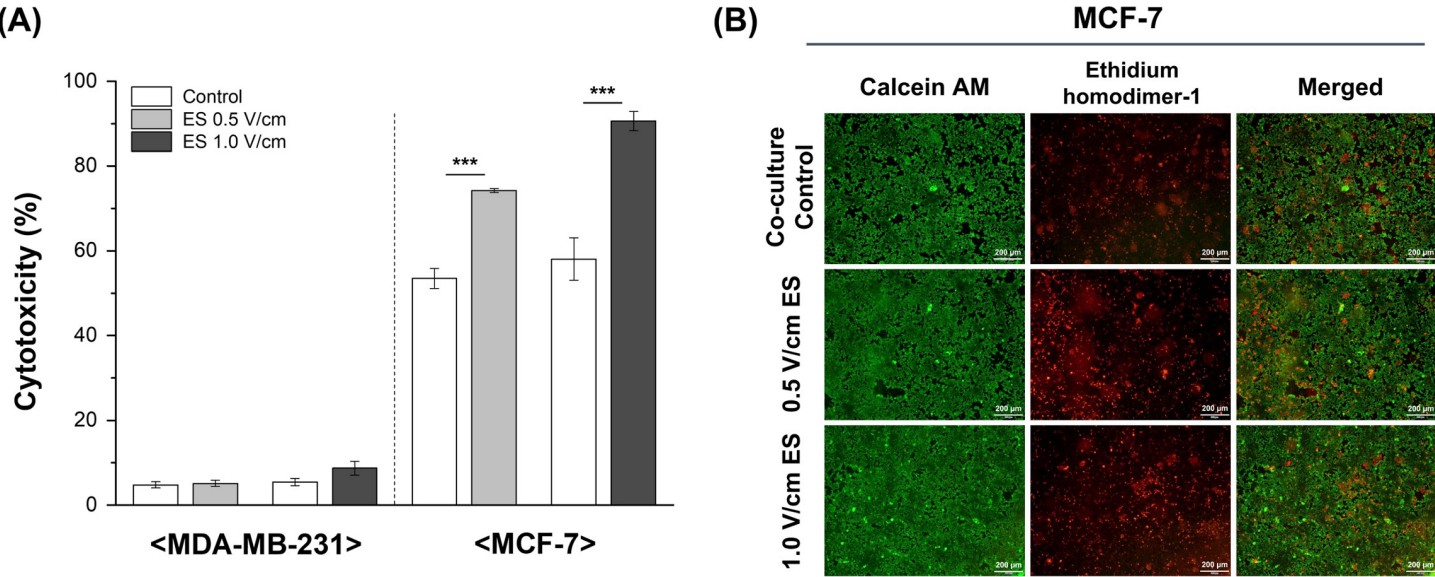

**Fig 3. Cytotoxicity of NK cells against tumor cells using LDH cytotoxicity assays.** (A) LDH cytotoxicity data of MDA-MB-231, actin response-positive cell line, and MCF-7, actin response-negative cell line treated for 24 h with electrically stimulated or non-stimulated KHYG-1 cells (n = 3). (B) Live/dead cytotoxicity imaging of MCF-7 cells co-cultured with stimulated or unstimulated KHYG-1 cells for 6 h. Results are presented with average and error bars. Student's t-test was used to determine the statistical significance of the results (*, P < 0.05; **, P < 0.01 and ***, P < 0.001).

cells was performed after co-culturing with stimulated or non-stimulated KHYG-1 cells for 6 h (Fig 3B). Therefore, our results showed that electrical stimulation improved the cytotoxic activity of NK cells against MCF-7 cells, whereas no significant changes were observed in MDA-MB-231 cells.

### Changes in gene and protein expression in NK cells following electrical stimulation

We confirmed the changes in the expression of genes related to the tumor-killing activity of KHYG-1 cells after exposure to electrical stimulation (Fig 4A). GZMB gene expression significantly increased at both 0.5 V/cm and 1.0 V/cm. Conversely, IFNG expression significantly decreased following exposure to electrical stimulation under both voltage conditions. TNFA gene expression significantly decreased following exposure to electrical stimulation at 1.0 V/cm.

ELISA was used to analyze granzyme B protein expression (Fig 4B). Upregulated Granzyme B expression leads to changes in protein expression. The intracellular levels of granzyme B in electrically stimulated KHYG-1 cells were high at 0.5 V/cm and significantly higher at 1.0 V/cm (1.26-fold change). Granzyme B secreted into the culture medium was sampled at 4 and 8 h after electrical stimulation. The extracellular level of granzyme B in the culture medium was 1.20 times higher than that in control after 4 h of incubation and 0.5 V/cm electrical stimulation. Therefore, the results showed that gene expression of granzyme B, a core effector molecule, was significantly upregulated after 0.5 V/cm and 1.0 V/cm electrical stimulations, which also led to an increase in protein levels.

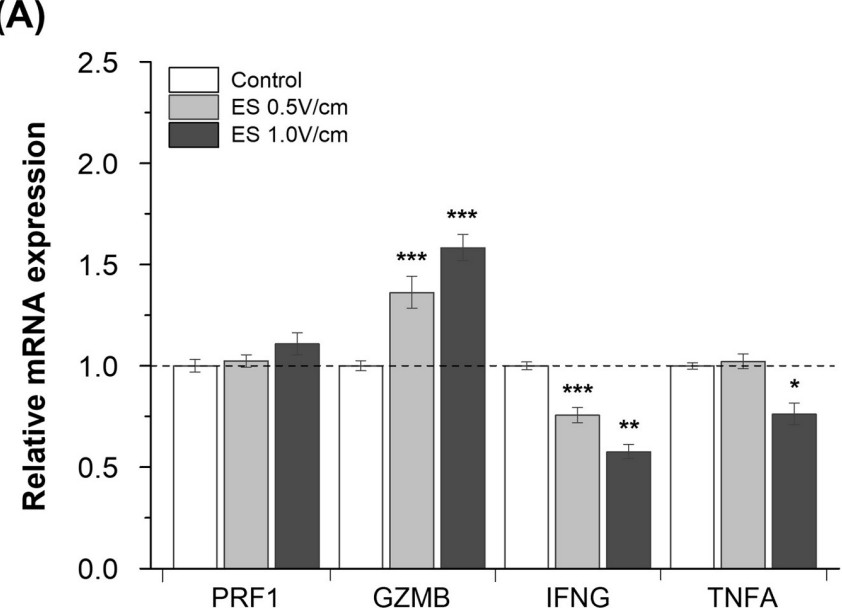

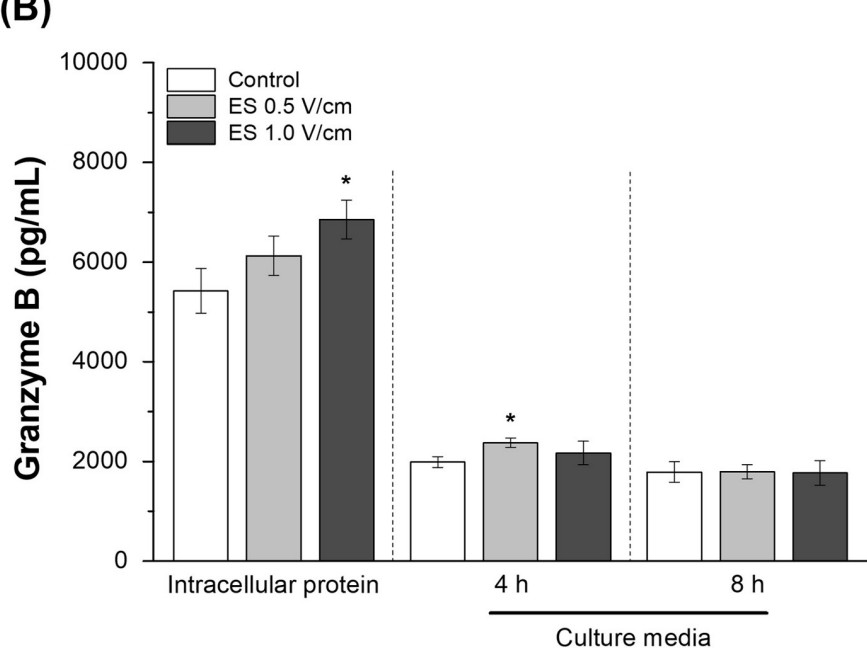

**Fig 4. Fold changes in gene expression levels related to cytotoxic activity and granzyme B production in NK cells after electrical stimulation.** (A) Relative mRNA expression levels of degranulation markers (PRF1, GZMB) and cytokines (IFNG, TNFA) in NK cells were analyzed and normalized to the housekeeping gene GAPDH and non-stimulated control (n = 3). (B) Granzyme B levels were detected in intracellular protein and extracellular levels were released into the culture media after incubation for 4 and 8 h by sandwich ELISA (n = 3). Results are presented with average and error bars. Student's t-test was used to determine the statistical significance of the results (*, P < 0.05; **, P < 0.01 and ***, P < 0.001).

### Changes in intracellular calcium concentration and calcium-mediated signaling

A Fluo-4 assay was performed to investigate the effect of electrical stimulation on the intracellular $Ca^{2+}$ levels affected by calcium influx (Fig 5A). The concentration of $Ca^{2+}$ was measured after 45 min of electrical stimulation, and the relative fluorescence level was calculated, which increased by 1.31-fold (0.5 V/cm) and 1.11-fold (1.0 V/cm) after electrical stimulation. In addition, the expression of calcium-mediated signaling proteins, which can be activated by $Ca^{2+}$ influx after electrical stimulation, was detected by RT-qPCR (Fig 5B). All calcium-related markers in the calcineurin-NFAT pathway, except IP3R and NFAT2, were significantly upregulated following stimulation with two different electrical voltages. NFAT2 expression also increased marginally after electrical stimulation. However, this increase was not statistically significant. In contrast, the IP3R expression level significantly decreased after electrical stimulation (Fig 5). This suggests that electrical stimulation affects $Ca^{2+}$ influx and downstream pathways via $Ca^{2+}$-dependent signaling proteins.

### Decrease in gene expression with calcium chelator BAPTA-AM

To further examine the effects of calcium signaling on NK cell degranulation, the cell-permeable calcium chelator, BAPTA-AM, was used. KHYG-1 cells were treated with 5 μM BAPTA-AM for 30 min before and 1 h after electrical stimulation. There was no significant difference in cell viability with 5 μM BAPTA-AM, with or without electrical stimulation (S1 Fig). The gene expression of GZMB and calcium signaling-related proteins (CALM, CALN, NFAT1, and CAMK II) was analyzed using RT-qPCR (Figs 6A and S2). By chelating free intracellular $Ca^{2+}$, GZMB expression did not increase significantly after electrical stimulation in the presence of the calcium chelator BAPTA-AM. Other markers associated with the calcineurin-NFAT signaling pathway showed a much lower increase in gene expression with BAPTA-AM than with electrical stimulation without BAPTA-AM (Figs 5B and 6A). The results showed that oscillations in intracellular $Ca^{2+}$ concentrations after electrical stimulation were suppressed via chelation, followed by the suppression of $Ca^{2+}$-dependent signaling.

### Increased dephosphorylation of NFAT1 with transcriptional activation by electrical stimulation

Western blot analysis of NFAT1 was conducted to determine the level of dephosphorylation of the transcription factor NFAT1 following exposure to electrical stimulation, which could lead to transcriptional activity. GAPDH (37 kDa) was used as a loading control. The upper band (140 kDa) represented phosphorylated NFAT1 (p-NFAT1), whereas the lower band (120 kDa) represented dephosphorylated NFAT1 (Fig 6B). NFAT1 dephosphorylation was quantitatively expressed as the ratio of NFAT1 to p-NFAT1 and calculated as a fold change relative to the control. Dephosphorylated NFAT1 levels increased 2.21-fold after electrical stimulation compared with the control, whereas treatment with the $Ca^{2+}$ chelator BAPTA-AM inhibited the effects of electrical stimulation on calcineurin-mediated dephosphorylation of NFAT1 with a 0.17-fold decrease.

## Discussion

Calcium signaling is a key mechanism in immune cells, from immunological synapse formation to lytic granule exocytosis. In particular, $Ca^{2+}$ influx through CRAC/ORAI channels triggered by immune activation contributes to $Ca^{2+}$ signaling and functions in immune cells [14–16]. Its importance in effector function has been widely recognized; however, the mechanisms

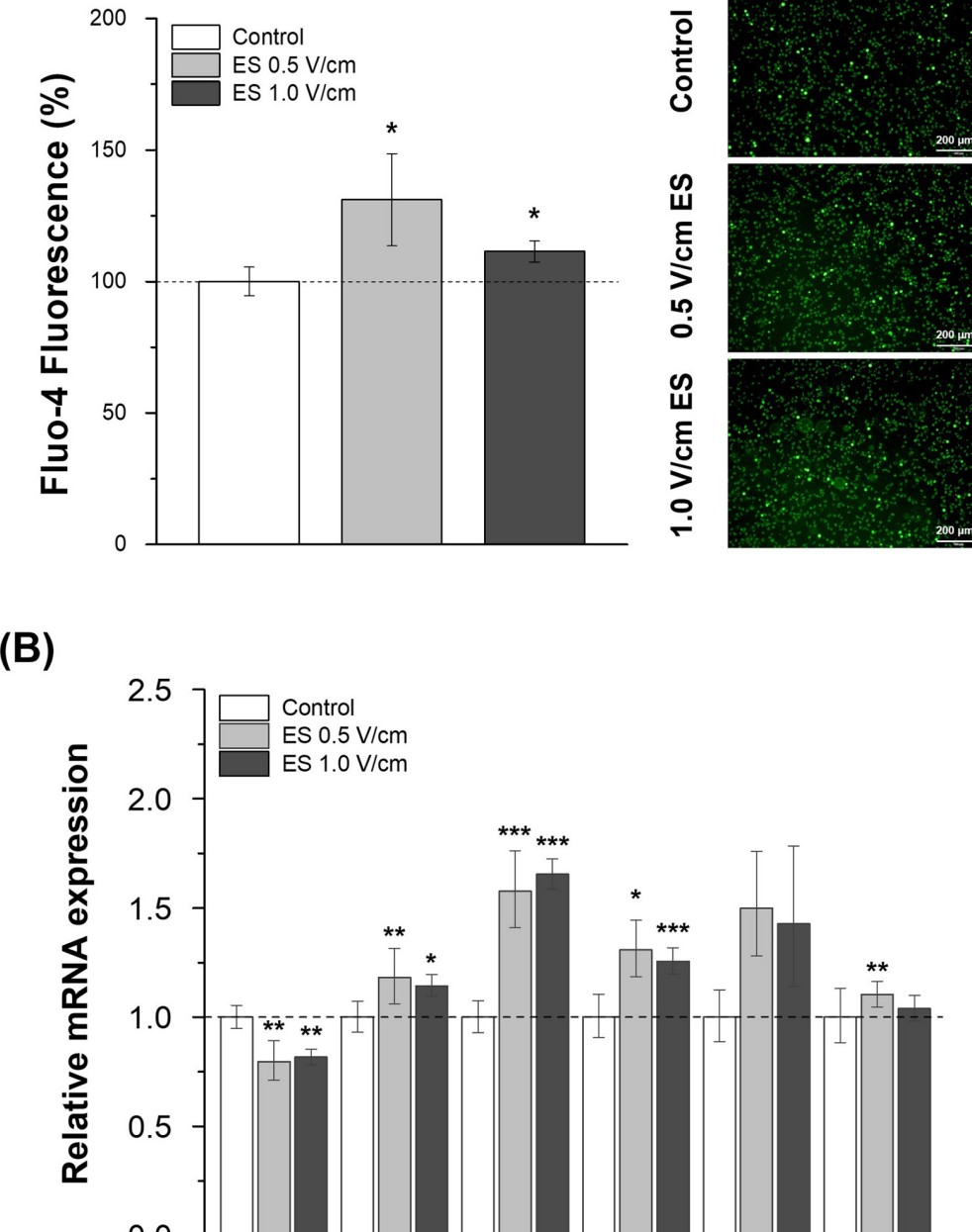

**Fig 5. Intracellular Ca²⁺ was measured by Fluo-4 fluorescence and relative gene expression levels involved in the Ca²⁺-NFAT pathway in response to electrical stimulation.** (A) Quantitative analysis of calcium with the Fluo-4 NW assay and fluorescence imaging of Fluo-4-stained cells was performed independently to detect $Ca^{2+}$ influx triggered by constant electrical stimulation (n = 3). (B) Gene expression levels of calcium signaling markers were analyzed using RT-qPCR after 1 h of stimulation (n = 3). Results are presented with average and error bars. Student's t-test was used to determine the statistical significance of the results (*, $P < 0.05$; **, $P < 0.01$ and ***, $P < 0.001$).

underlying this enhancement in NK cells are not well characterized. In this regard, this study applied an electrical stimulation system as a method for enhancing the cytotoxic activity of NK

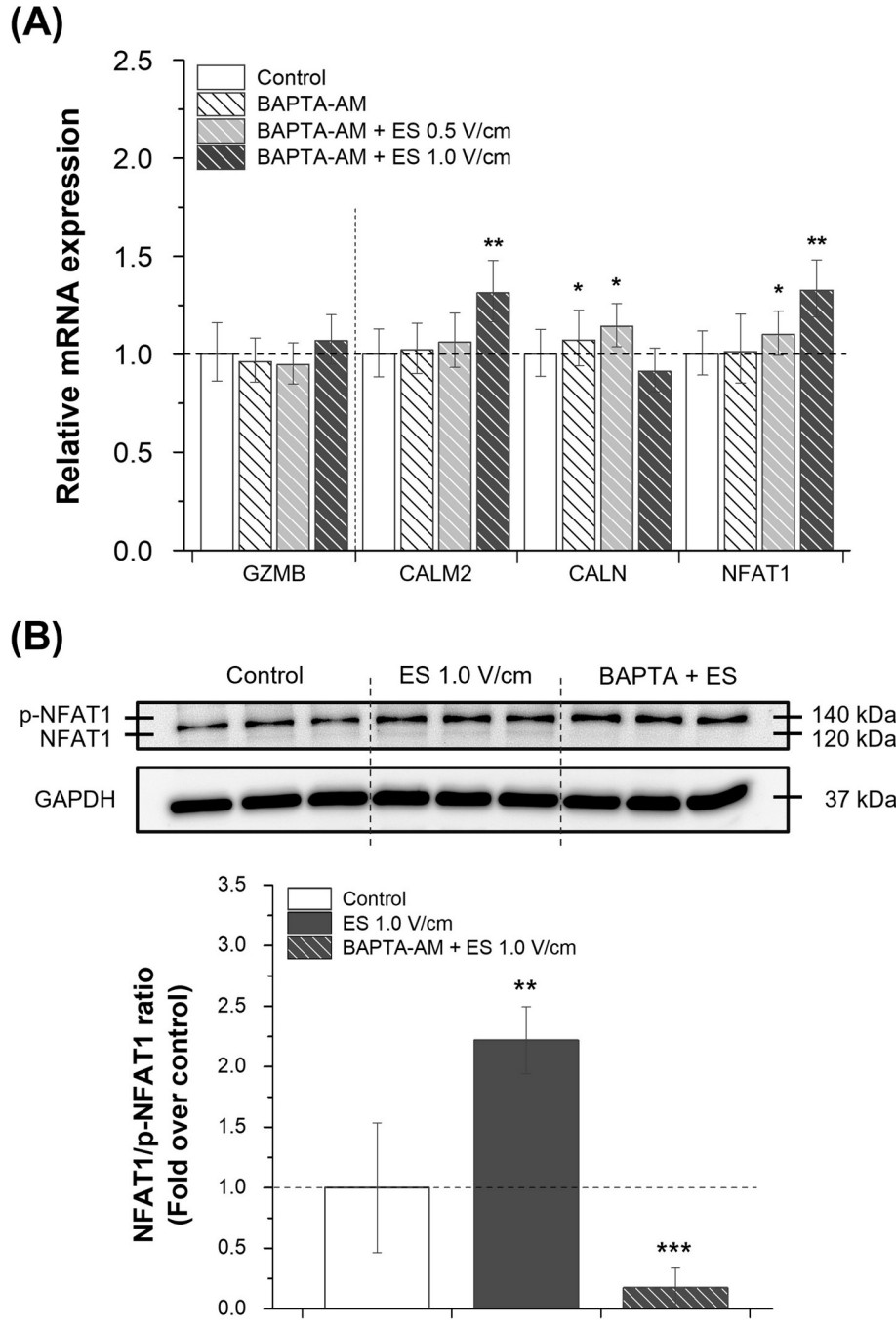

**Fig 6. Effects of electrical stimulation-induced increase in calcium levels on gene expression and NFAT1 dephosphorylation with the calcium chelator BAPTA-AM.** (A) RT-qPCR was performed to analyze GZMB expression levels and calcineurin-NFAT signaling proteins (CALM, CALN, and NFAT1) after $Ca^{2+}$ chelation. KHYG-1 cells were cultured with DMSO (control) or a calcium chelator (5 μM BAPTA-AM), with or without electrical stimulation (n = 3). (B) Western blot and densitometric analyses were conducted under the same conditions (1.0 V/cm) to determine the dephosphorylation of NFAT1, which could represent the transcriptional activation in the $Ca^{2+}$-dependent pathway. Quantification of dephosphorylated NFAT1 was calculated as the NFAT1/p-NFAT1 ratio relative to the control (n = 3). Results are presented with average and error bars. Student's t-test was used to determine the statistical significance of the results (\*, P < 0.05; \*\*, P < 0.01 and \*\*\*, P < 0.001).

cells and focused on the involvement and influence of intracellular $Ca^{2+}$ in the efficient effector function of immune cells.

The conditions for electrical stimulation were based on previous studies (Fig 1A) [21, 22]. Electrical stimulation with two different DC voltages was provided continuously for 1 h without interruption. In comparison with the control, the relatively low BAX to BCL-2 ratio with decreased pro-apoptotic BAX and upregulated anti-apoptotic BCL-2 showed that electrical stimulation did not induce apoptosis, regardless of the voltage conditions used (Fig 2B) [23].

In the LDH cytotoxicity assays, changes in cytotoxicity against the MCF-7 cell line co-cultured with electrically stimulated NK cells appeared to be a cell line-specific response. The NK cell-mediated cytotoxicity of actin response-negative MCF-7 cells increased with electrically stimulated NK cells owing to their high susceptibility to NK cell-induced lysis but not in actin response-positive MDA-MB-231 cells [8]. To determine the mechanism by which electrical stimulation influences the increase in cytotoxic activity against MCF-7 cells, changes in the gene expression of related lytic granules and cytokines secreted from NK cells were evaluated (Fig 4). These results indicated that higher granzyme B expression and degranulation resulted in a more potent effector function via perforin-dependent uptake of granzyme B [13, 24]. In this process, the expression and degradation of secretory lysosomes are often accompanied by an increase in $Ca^{2+}$ levels. Cytosolic $Ca^{2+}$ concentration plays an important role in immune cells such as cytotoxic T cells and NK cells. Electrical stimulation alters $Ca^{2+}$ influx and gene expression by mediating calcium signaling in various cell lines [25–27]. Based on these previous studies and experimental results, a further assay was conducted on $Ca^{2+}$ to determine whether $Ca^{2+}$ flux was related to the killing ability mediated by the degranulation of lytic granules in immune cells. The results of the Fluo-4 assay confirmed that stimulating cells in the electric field triggered an increase in intracellular $Ca^{2+}$ (Fig 5A), possibly mediated by calcium channels [28], followed by activation of the calcineurin-NFAT signaling pathway (Fig 7), which was analyzed using RT-qPCR (Fig 5B). The results of this study suggested that a minor influx of $Ca^{2+}$ (through electrical stimulation) had a positive effect on the anti-tumor cytotoxicity of NK cells by affecting transcriptional regulation through $Ca^{2+}$ signaling (Fig 7) [16, 29].

To implicate $Ca^{2+}$ and downstream signals as the cause of electrical stimulation-mediated transcriptional changes, an experiment was conducted using 5 μM BAPTA-AM [30], a $Ca^{2+}$ chelator. Changes in the expression of granzyme B and $Ca^{2+}$-dependent signaling proteins were analyzed (Fig 6A). It was confirmed that oscillations in intracellular $Ca^{2+}$ concentrations after electrical stimulation were suppressed through chelation, followed by the suppression of $Ca^{2+}$-dependent signaling. In addition, it was confirmed that $Ca^{2+}$ was involved in the process from biogenesis to degranulation of granules, considering that GZMB expression was suppressed by treatment with a $Ca^{2+}$ chelator (Fig 6A). The expression levels of CAMK II and NFAT2 engaged in calcium signaling were also increased by electrical stimulation. However, the absolute expression levels of CAMK II (S2 Fig) and NFAT2 (Fig 5B) in KHYG-1 cells were low and unstable (fluctuating). Therefore, they were excluded as markers in the chelating experiment (Fig 6A). Although it was not a marker of particular interest within the experiment, the expression of NFAT2, a member of the NFAT family, was also found to be an important regulator of granzyme B-mediated cytotoxicity in T cells [31]. Thus, it is necessary to study $Ca^{2+}$ signaling in cytotoxic immune cells.

NFAT1 dephosphorylation, mediated by the protein phosphatase calcineurin, was preceded by nuclear translocation, DNA-binding activity, and transcriptional regulation of gene expression (Fig 7) [32]. According to the western blotting results (Fig 6B), electrically stimulated (1.0 V/cm) KHYG-1 cells exhibited a 2.21-fold higher level of dephosphorylated NFAT1 than the control. This indicates that electrical stimulation induced the transcriptional activity of NFAT1 with $Ca^{2+}$ influx and further transcriptional regulation of inducible target genes, such

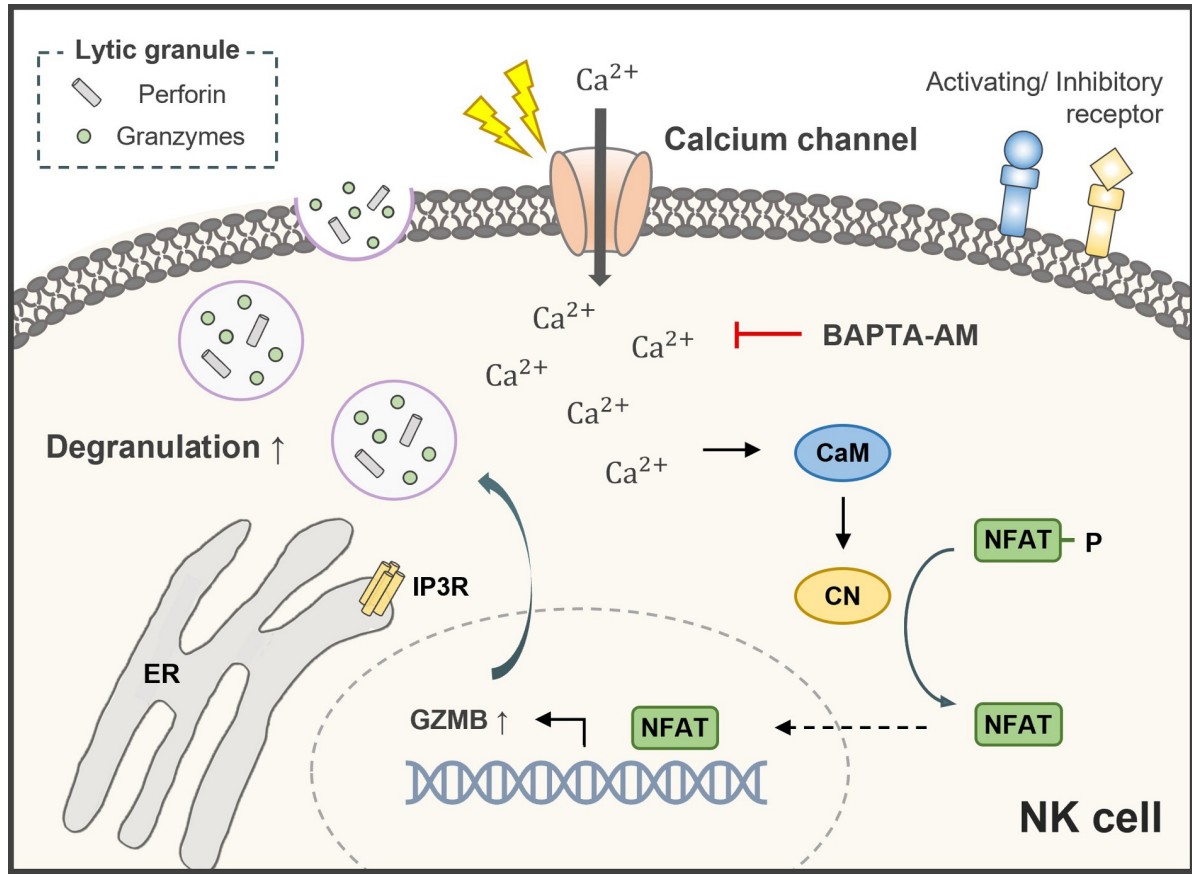

**Fig 7. Calcineurin-NFAT signaling pathway.** Calcium-mediated signaling is involved in the activation of diverse subsets of gene transcription, including GZMB. Target gene expression can change with the activation of transcriptional factors in response to electrical stimulation, thus providing a link between intracellular $Ca^{2+}$ and gene expression in immune effector functions. ER, endoplasmic reticulum; IP3R, inositol 1,4,5-trisphosphate receptor; CaM, calmodulin; CN, calcineurin; NFAT, nuclear factor of activated T cells; GZMB, granzyme B.

as granzyme B. Dephosphorylation was inhibited by BAPTA-AM by 0.17-fold, indicating that the electrical stimulation-induced alteration was mediated by the $Ca^{2+}$ signaling pathway, resulting in changes in the transcription of the related genes. In addition, this demonstrates that one of the roles of the transcription factor NFAT1 in immune cells is to regulate the expression of cytolytic protein genes.

Therefore, our study reveals a positive effect of electrical stimulation on the cytotoxicity of NK cells by controlling intracellular $Ca^{2+}$ influx and mediating signaling without any genetic manipulation. This process was accomplished by activating calcineurin-NFAT signaling (Fig 7).

## Conclusions

Numerous studies have been conducted to define the mechanism of cytotoxicity regulation and improve the cytotoxicity of immune cells. Further elucidation of the correlation between $Ca^{2+}$ flux and enhanced cytotoxicity following electrical stimulation will provide valuable insights for NK cell application in anti-tumor treatments.

In this study, we observed that a minor increase in electrical stimulation-induced $Ca^{2+}$ signaling affected the biogenesis of lytic granules in NK cells by upregulating granzyme B expression. We also found that the transcription factor NFAT1 may be involved in regulating granzyme B

expression via Ca$^{2+}$ and calcineurin-dependent mechanisms. Engineering cells with electrical stimulation for increased granzyme B accumulation and enhanced cytolytic degranulation may provide a novel perspective for developing anti-tumor treatments, thereby expanding the therapeutic potential alone or in combination with existing adoptive or targeted cell therapies.

## Supporting information

**S1 Fig. Effects of electrical stimulation and BAPTA-AM on NK cell viability.** (A) LDH cytotoxicity assay after 24 h of incubation following electrical stimulation (1 h) (n = 3). Relative LDH release levels were calculated relative to the control group in terms of fold change. CCK-8 assays were used to detect (B) the long-term effects of electrical stimulation on cell viability after 24 h or 48 h and (C) the effects of BATPA-AM or electrical stimulation with BAPTA-AM on the viability of NK cells (n = 3).
(TIF)

**S2 Fig. Effects of electrical stimulation and BAPTA-AM on calcium signaling.** Relative gene expressions were normalized to GAPDH and expressed as fold change over the control group. One-way ANOVA and Tukey's post-hoc test (R Studio program, http://www.rstudio.com/) were conducted for multiple comparisons (n = 3, * or #, P < 0.05; ** or ##, P < 0.01 and *** or ###, P < 0.001).
(TIF)

**S3 Fig. Full images of western blots presented in the manuscript.** Full-length images of NFAT1, Lamin B as a nuclear loading control, and GAPDH as a loading control. Whole-cell protein sample prepared by RIPA lysis buffer (A)–(F), and nuclear extraction kit (ab113474, Abcam) (G)–(L); (A), (G) western blotting full images with marker; and (B), (C) and (H), (I) inverted, overexposed version of each full image; (E), (K) full-length image; (F), (L) overexposed version of full-length images; (D), (J) cropped version similar to that of Fig 5B presented in the manuscript.
(TIF)

## Author Contributions

**Conceptualization:** Minseon Lee, Soonjo Kwon.

**Data curation:** Minseon Lee, Soonjo Kwon.

**Formal analysis:** Minseon Lee, Soonjo Kwon.

**Funding acquisition:** Soonjo Kwon.

**Investigation:** Minseon Lee, Soonjo Kwon.

**Methodology:** Minseon Lee, Soonjo Kwon.

**Project administration:** Minseon Lee, Soonjo Kwon.

**Resources:** Minseon Lee, Soonjo Kwon.

**Supervision:** Soonjo Kwon.

**Validation:** Minseon Lee, Soonjo Kwon.

**Visualization:** Minseon Lee, Soonjo Kwon.

**Writing – original draft:** Minseon Lee.

**Writing – review & editing:** Soonjo Kwon.

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
