## [Decision Letter · Decision Letter 0]

16 Oct 2023

PONE-D-23-27697Enhanced cytotoxic activity of natural killer cells from increased calcium influx induced by electrical stimulationPLOS ONE

Dear Dr. Kwon,

Thank you for submitting your manuscript to PLOS ONE. After careful consideration, we feel that it has merit but does not fully meet PLOS ONE’s publication criteria as it currently stands. Therefore, we invite you to submit a revised version of the manuscript that addresses the points raised during the review process.

The reviewers have provided detailed comments on your manuscript, highlighting both its strengths and areas that require improvement. I strongly encourage you to carefully review their feedback, addressing each point to enhance the quality and clarity of your work. Please find the attached reviewers' comments and suggested revisions below.

It is essential that you pay particular attention to the major revisions identified by the reviewers, as these changes are crucial to meet the journal's standards. You should also provide a detailed response to each comment, explaining how you have addressed the concerns and where applicable, indicating the page numbers in your revised manuscript where the changes have been made.

Please submit your revised manuscript within Nov 30 2023 11:59PM. If you will need more time than this to complete your revisions, please reply to this message or contact the journal office at plosone@plos.org. Please include the following items when submitting your revised manuscript:A rebuttal letter that responds to each point raised by the academic editor and reviewer(s). You should upload this letter as a separate file labeled 'Response to Reviewers'.A marked-up copy of your manuscript that highlights changes made to the original version. You should upload this as a separate file labeled 'Revised Manuscript with Track Changes'.An unmarked version of your revised paper without tracked changes. You should upload this as a separate file labeled 'Manuscript'.

We look forward to receiving your revised manuscript.

Kind regards,

Charushila Yuvraj Yuvraj Kadam, PhD

Academic Editor

PLOS ONE

This research was supported by a National Research Foundation of Korea (NRF) grant funded by the Korea government (MSIT) (RS-2023-00207801) and an Inha University Research Grant, Korea. 

NO - Include this sentence at the end of your statement: The funders had no role in study design, data collection and analysis, decision to publish, or preparation of the manuscript.

Soonjo Kwon received the following fund.

Reviewers' comments:

Reviewer's Responses to Questions

**Comments to the Author**

1. Is the manuscript technically sound, and do the data support the conclusions?

Reviewer #1: Partly

Reviewer #2: Partly

2. Has the statistical analysis been performed appropriately and rigorously? 

Reviewer #1: Yes

Reviewer #2: Yes

3. Have the authors made all data underlying the findings in their manuscript fully available?

Reviewer #1: Yes

Reviewer #2: Yes

4. Is the manuscript presented in an intelligible fashion and written in standard English?

Reviewer #1: No

Reviewer #2: Yes

5. Review Comments to the Author

Reviewer #1: 1. Can you provide more information on the specific electrical stimulation system used in the study? How was it designed and implemented to ensure accurate and controlled electrical stimulation of the NK cells?

2. How was the correlation between elevated intracellular calcium levels and NK cell activation determined? Were any other factors considered or controlled for in this analysis?

3. You mentioned the cytotoxicity of electrically stimulated KHYG-1 cells against breast cancer MCF-7 cells. Were other tumor cell lines tested, or was the experiment limited to MCF-7 cells? If other cell lines were used, were there any noticeable differences in cytotoxicity?

4. The study highlights the gene expression of granzyme B as a key factor in the increased cytotoxicity of electrically stimulated NK cells. Were there any other genes or proteins that showed significant changes in expression after electrical stimulation?

5. How specific is the role of calcium influx in mediating the effects of electrical stimulation on NK cell function? Were there any other signaling pathways or factors that were explored in relation to NK cell activation and cytotoxicity?

6. The study focuses on breast cancer cells, but could the findings be applicable to other types of cancer as well? Is there any indication or speculation about the potential broader implications of this research in terms of cancer immunotherapy?

7. Were there any potential limitations or challenges encountered during the study that may have influenced the results? Are there any aspects of the experimental design or methodology that should be taken into consideration when interpreting the findings?

8. Based on the findings of this study, what are the potential next steps or future research directions in exploring the use of electrical stimulation for enhancing NK cell function in immunotherapy? Are there any specific areas that warrant further investigation?

Reviewer #2: The authors investigated a new method for improving the functional potential of NK cells through electrical stimulation. This is a well written and structured research article. However, there are several issues with the results and discussion that need to be clarified/addressed. Below are more specific comments by section:

Materials and methods

Western blotting:

The authors should clarify why 1.0 V/cm stimulation was chosen or be consistent and provide results for both 0.5 and 1.0 V/cm electrical stimulation.

Results

Effects of electrical stimulation on cell viability of NK cells:

The authors examine only viability, but not the ability to proliferate. For therapeutic use, NK cell proliferation is one of the most important factors, and adding this information to the article can be very valuable.

Effects of electrical stimulation on the cytotoxic activity of NK cells:

I suppose this is just a typo: “The MDA-MB-231 cells electrically stimulated with KHYG-1 showed no significant 220 increase in cytotoxicity” and it should be “The MDA-MB-231 cells with electrically stimulated KHYG-1 showed no significant 220 increase in cytotoxicity”.

In this section, two co-culture times and two methods were mentioned: 24 hours and 6 hours. The authors should clarify this section to avoid confusion since it is not obvious what the reader should understand from Figure 3B and why 6 hours was used.

Fluo-4 calcium fluorescence assay:

The incubation time with Flow-4 AM was 45 min. This is confusing because the duration of the other experiments was 1 hour. If the authors can provide results with multiple time points such as 15, 30, 45, 60 minutes, it will increase the scientific value of this work, otherwise the authors should be consistent and use the same time for all experiments.

Discussion

The difference in results between the 0.5 and 1.0 V/cm groups needs to be clarified.

The authors should discuss why Figure 5a shows that Ca2+ levels are higher for the 0.5 V/cm group and cytotoxicity is higher for the 1 V/cm group.

For the concentration of granzyme B intracellular concentration is higher for 1 V/cm but for culture media the situation is reverse at 4h.

The concentration of granzyme B after 8 hours is the same for all three groups. This fact may lead to the conclusion that electrical stimulation has a short-term effect and cannot be considered as a potential treatment, and all differences can be explained by the effect within the first 4 hours and the fact that MCF-7 cell line is susceptible for the KHYG-1 cells that is highly cytotoxic cell line [1]

Another question is why 1h co-incubation was selected, what would happened with different incubation time?

[1] G. Suck,

---

## [Author Response · Author response to Decision Letter 0]

22 Jan 2024

November 27, 2023

We thank the reviewers for the insightful and constructive suggestions. We are addressing them below. The revised manuscript has been much improved. The reviewers’ comments are in italic, while responses are in plain text.

Reviewer #1’s comments:

1) Can you provide more information on the specific electrical stimulation system used in the study? How was it designed and implemented to ensure accurate and controlled electrical stimulation of the NK cells?

Response: As described in Fig 1A of Materials and Methods section, the electrical stimulation system consisted of two platinum electrodes (cathodes and anodes, 20 mm apart in parallel) on a lid above each well and a function generator [1–4]. The condition of stimulation was controlled by the function generator, and its performance was verified using an oscilloscope. It was designed to affect cells directly through the culture medium by inserting electrodes, which were fully applied throughout the well.

We revised the corresponding sentences at the line 86~92 in Page 5.

“ ~ The electrical system chamber was printed using a 3D printer (3DP-110F; Cubicon, Seongnam, Korea) fitted with a lid for standard 6-well plates (Fig 1A). The electrical stimulation system consisted of two platinum (Pt) electrodes (cathodes and anodes) (99.9% Pt wire with 0.3 mm diameter) at a distance of 20 mm in parallel (Fig 1). The condition of stimulation was controlled by the function generator. It was designed to affect cells directly through the culture medium by inserting electrodes, which were fully applied throughout the well. ~”

2) How was the correlation between elevated intracellular calcium levels and NK cell activation determined? Were any other factors considered or controlled for in this analysis?

Response: Previous studies have reported that applied electrical stimulation could induce changes in cell differentiation, proliferation, and migration, and the effects were mediated by the calcium ion [1–4]. In this study, the stimulation could induce an increased level of gene and protein expression of granzyme B (RT-qPCR, ELISA data), with an increase in intracellular calcium level (Fig. 5A) and an upregulation in calcium signaling-related proteins (Fig. 5B). As we conducted the experiments with the calcium ion chelator BAPTA-AM, the correlation between elevated intracellular calcium level and NK cells’ degranulation seemed to be clearer (Fig. 6). Other factors except electrical stimulation were controlled, such as cell density, culture medium (type or total volume), and even incubation or culture time.

The related sentences are at the line 63~92 in Page 4.

“~ In previous studies, electrical stimulation has been used to regulate cellular functions such as proliferation, migration, and differentiation of mesenchymal stem cells (MSCs) by modulating Ca2+ entry and activating calcineurin-NFAT signaling. ~”

3) You mentioned the cytotoxicity of electrically stimulated KHYG-1 cells against breast cancer MCF-7 cells. Were other tumor cell lines tested, or was the experiment limited to MCF-7 cells? If other cell lines were used, were there any noticeable differences in cytotoxicity?

Response: In this study, we analyzed the cytotoxicity of stimulated NK cells against two different breast cancer cell lines such as MCF-7 (actin response-negative) and MDA-MB-231 (actin response-positive) [5,6] . Electrical stimulated KHYG-1 cells showed 1.27-fold (0.5 V/cm) and 1.55-fold (1.0 V/cm) higher cytotoxicity against MCF-7, while the increase of cytotoxicity against MDA-MB-231 was not statistically significant. Even in the case of the breast cancer cells, it varies depending on the nature of the cells. For the different types of cancer cells, the degree of electrical stimulated KHYG-1 cells’ cytotoxic effects may vary depending on the characteristics of the target cancer cells. However, these results suggested that it would be worth to test the therapeutic potential against other cancer cells in further studies. 

The related sentences are at the line 339~343 in Page 16.

“~ In the LDH cytotoxicity assays, changes in cytotoxicity against the MCF-7 cell line co-cultured with electrically stimulated NK cells appeared to be a cell line-specific response. The NK cell-mediated cytotoxicity of actin response-negative MCF-7 cells increased with electrically stimulated NK cells owing to their high susceptibility to NK cell-induced lysis but not in actin response-positive MDA-MB-231 cells.~”

4) The study highlights the gene expression of granzyme B as a key factor in the increased cytotoxicity of electrically stimulated NK cells. Were there any other genes or proteins that showed significant changes in expression after electrical stimulation?

Response: Our results point out that electrical stimulation has a significant impact on granzyme B gene expression, with 1.36-fold and 1.58-fold increases, and following consequent cytotoxicity. Expression for other cytotoxicity factors such as perforin, IFN-�, TNF-� was analyzed, but was not significantly changed by electrical stimulation. In addition to the findings of this research, the expression of genes related to cell cytoskeleton remodeling (ACTB, CDC42) seemed to be upregulated after electrical stimulation (data not shown). This phenomenon could be studied further since it was reported that actin network and cytoskeletal structure remodeling could also play a critical role in immune synapse with tumors and release of cytolytic granules [6–8].

5) How specific is the role of calcium influx in mediating the effects of electrical stimulation on NK cell function? Were there any other signaling pathways or factors that were explored in relation to NK cell activation and cytotoxicity?

Response: Previous studies have reported that the role of calcium ions was important in immune cells related to its cytotoxic function and activity (especially to degranulation). As NFAT-mediated pathway is known as one of the major transcriptional regulators of activation in immune cells, there are other factors-mediated pathways such as Janus kinases- signal transducer and activator of transcription proteins (JAK-STAT) signaling and nuclear factor kappa-light-chain-enhancer of activated B cells (NF-κB) pathway [9,10]. NFAT-driven gene expression and immune cell immunity were highly dependent on sustained Ca2+ influx and calcineurin activity. Others, on the other hand, were less dependent on Ca2+ ions and calmodulin/calcineurin signaling pathway which was focused in this study with electrical stimulation [3,11–13]. In this respect, the results of this study also could support the association between intracellular Ca2+-related factors (especially in NFAT pathway) and cytolytic activity of NK cells with degranulation.

The related sentences are at the line 347~358 in Pages 16 ~ 17.

“~ In this process, the expression and degradation of secretory lysosomes are often accompanied by an increase in Ca2+ levels. Cytosolic Ca2+ concentration plays an important role in immune cells such as cytotoxic T cells and NK cells. Electrical stimulation alters Ca2+ influx and gene expression by mediating calcium signaling in various cell lines [25–27]. Based on these previous studies and experimental results, a further assay was conducted on Ca2+ to determine whether Ca2+ flux was related to the killing ability mediated by the degranulation of lytic granules in immune cells. The results of the Fluo-4 assay confirmed that stimulating cells in the electric field triggered an increase in intracellular Ca2+ (Fig 5A), possibly mediated by calcium channels [28], followed by activation of the calcineurin-NFAT signaling pathway (Fig 7), which was analyzed using RT-qPCR (Fig 5B). The results of this study suggested that a minor influx of Ca2+ (through electrical stimulation) had a positive effect on the anti-tumor cytotoxicity of NK cells by affecting transcriptional regulation through Ca2+ signaling (Fig 7). ~”

6) The study focuses on breast cancer cells, but could the findings be applicable to other types of cancer as well? Is there any indication or speculation about the potential broader implications of this research in terms of cancer immunotherapy?

Response: We evaluated the cytotoxicity of NK cells against two breast cell lines MCF-7 and more aggressive MDA-MB-231 cells, which showed the significantly different response to immunotherapy in previous studies [5,6]. As speculated with the results against MCF-7 cells in this study (about 53~90% cell cytotoxicity, Fig 3), it could be sufficiently applicable to leukemia and even other type of solid tumors.

The related sentences are at the line 225~228 in Page 11.

“~ The MDA-MB-231 cells with electrically stimulated KHYG-1 showed no significant increase in cytotoxicity (Fig 3A). However, the cytotoxicity of MCF-7 with electrically stimulated KHYG-1 cells increased from 53% to 74% (1.27-fold, 0.5 V/cm electrical stimulation) and from 58% to 90% (1.55-fold, 1.0 V/cm electrical stimulation), respectively. ~”

7) Were there any potential limitations or challenges encountered during the study that may have influenced the results? Are there any aspects of the experimental design or methodology that should be taken into consideration when interpreting the findings?

Response: It was not able to evaluate the real-time changes in intracellular calcium level while exposed to electrical stimulation. Although the result confirmed by the alternative method (analyzed at 45 min timepoint after stimulation) would not affect the results and interpretation of our findings, it was the challenges encountered during the study.

8) Based on the findings of this study, what are the potential next steps or future research directions in exploring the use of electrical stimulation for enhancing NK cell function in immunotherapy? Are there any specific areas that warrant further investigation?

Response: As the electrical stimulation have been a potential tool in tissue engineering, especially being efficient to stem cell proliferation and differentiation [2,14,15]. In immunotherapy in which area related to this study, further studies can be able to use this tool for cell culture inducing cell proliferation, activation, and cell phenotype by stimulating Ca2+-related transcriptional factor such as NFAT. Furthermore, it can also be applied to modulate immune system, enhancing immune cell proliferation, secretion of cytokines and degranulation [16,17].

The related sentences are at the line 59~72 in Pages 3~4.

Reviewer #2’ s comments:

The authors investigated a new method for improving the functional potential of NK cells through electrical stimulation. This is a well written and structured research article. However, there are several issues with the results and discussion that need to be clarified/addressed. Below are more specific comments by section:

1) Materials and methods. Western blotting: The authors should clarify why 1.0 V/cm stimulation was chosen or be consistent and provide results for both 0.5 and 1.0 V/cm electrical stimulation.

Response: We performed NFAT1 Western blot analysis using 1.0 V/cm stimulation samples, which are representative of stimulation conditions (possible maximum in this study), taking into account the results of cytotoxicity, granzyme B expression levels, and calcium signaling-related protein expression. Changes in granzyme B expression level and protein expression related to calcium signaling were observed in the range (0.5 ~ 1.0 V/cm) that did not affect cell viability. Similar response was described at #5) response. 

We revised the corresponding sentences at the line 381~386 in Pages 18.

“ ~ NFAT1 dephosphorylation, mediated by the protein phosphatase calcineurin, was preceded by nuclear translocation, DNA-binding activity, and transcriptional regulation of gene expression (Fig 7). According to the western blotting results (Fig 6B), electrically stimulated (1.0 V/cm) KHYG-1 cells exhibited a 2.21-fold higher level of dephosphorylated NFAT1 than the control. This indicates that electrical stimulation induced the transcriptional activity of NFAT1 with Ca2+ influx and further transcriptional regulation of inducible target genes, such as granzyme B. ~” 

2) Results. Effects of electrical stimulation on cell viability of NK cells: The authors examine only viability, but not the ability to proliferate. For therapeutic use, NK cell proliferation is one of the most important factors, and adding this information to the article can be very valuable.

Response: We evaluated the effect of electrical stimulation on cell viability with CCK-8 assay which could reflect mitochondrial activity, LDH release level, and BAX/BCL2 gene expression ratio at 24 and 48 hours after the stimulation. But as you suggested, we revised the Results section with the relevant information (lines 196-198), and the related data has been also included as S1 Fig. The CCK-8 assay results of NK cells at 24 h and 48 h after stimulation was over the 95 % of viability compared to control group (S1 fig). This result could suggest that there was no long-term negative effect of applied stimulation on cell viability of NK cells, and also no negative effects on the ability to proliferate.

3) Results. Effects of electrical stimulation on the cytotoxic activity of NK cells: I suppose this is just a typo: “The MDA-MB-231 cells electrically stimulated with KHYG-1 showed no significant increase in cytotoxicity” and it should be “The MDA-MB-231 cells with electrically stimulated KHYG-1 showed no significant increase in cytotoxicity”. In this section, two co-culture times and two methods were mentioned: 24 hours and 6 hours. The authors should clarify this section to avoid confusion since it is not obvious what the reader should understand from Figure 3B and why 6 hours was used.

Response: We apologized for some errors the reviewer indicated (lines 225-226). We revised the Results section and also double-checked similar mistakes throughout the manuscript. The cytotoxicity of NK cells against cancers was evaluated at 24 h of co-culture time. The live/dead cytotoxicity imaging was conducted at 6 h timepoint during co-culture because dead MCF-7 cells can be detached at the end of co-culture (24 h), which could make the results uncertain. As you suggested, we revised the Materials and Methods section, and the relevant information has been included.

We also revised the corresponding sentences at the line 120~122 in Pages 6.

“ ~ A live/dead cytotoxicity assay (L3224; Invitrogen) was used to measure the NK cell cytotoxicity at 6 h timepoint during co-culture, because most dead MCF-7 cells can be detached 24 hours after co-culture. ~”

4) Results. Fluo-4 calcium fluorescence assay: The incubation time with Flow-4 AM was 45 min. This is confusing because the duration of the other experiments was 1 hour. If the authors can provide results with multiple time points such as 15, 30, 45, 60 minutes, it will increase the scientific value of this work, otherwise the authors should be consistent and use the same time for all experiments.

Response: According to the manufacturer’s protocol, the incubation time for Fluo-4 AM dye staining at 37 °C was up to 45 minutes. Following the recommended instruction, we carried out with maximum staining time while carrying electrical stimulation. The calcium assay for multiple timepoint could not be conducted simultaneously because of the condition with stimulation system. Therefore, calcium fluorescence assay was conducted only with 45 minutes also with consideration of plate-to-plate variation of each timepoint. We added additional explanations in Materials and Methods section of current manuscript.

We revised the corresponding sentences at the line 157~159 in Pages 8.

“~ KHYG-1 cells were incubated with Fluo-4 AM dye for 45 min at 37 °C while being exposed to each voltage range of electrical stimulation, which was performed during the maximum time recommended in the manufacturer’s protocol for staining dye.~”

5) Discussion. The difference in results between the 0.5 and 1.0 V/cm groups needs to be clarified. The authors should discuss why Figure 5a shows that Ca2+ levels are higher for the 0.5 V/cm group and cytotoxicity is higher for the 1 V/cm group.

---

## [Decision Letter · Decision Letter 1]

3 Apr 2024

Enhanced cytotoxic activity of natural killer cells from increased calcium influx induced by electrical stimulation

PONE-D-23-27697R1

Dear Dr. Soonjo Kwon,

We’re pleased to inform you that your manuscript has been judged scientifically suitable for publication and will be formally accepted for publication once it meets all outstanding technical requirements.

Kind regards,

Charushila Yuvraj Yuvraj Kadam, PhD

Academic Editor

PLOS ONE

Additional Editor Comments (optional): 

Upon careful consideration of the revised manuscript and the responses provided by the authors to the revisions suggested by Reviewer 1, I determined that the paper possesses significant academic merit deserving of publication. While Reviewer 1 recommended rejection, Reviewer 2 expressed support for publication, and I found that the revisions adequately addressed the concerns raised by Reviewer 1.

My decision to accept the manuscript despite the recommendation for rejection was based on a holistic evaluation of the paper's content, originality, and potential contribution to the field. I believe that the strengths of the research outweigh the concerns raised by Reviewer 1, and therefore, I decided to recommend acceptance.

Reviewers' comments:

Reviewer's Responses to Questions

**Comments to the Author**

1. If the authors have adequately addressed your comments raised in a previous round of review and you feel that this manuscript is now acceptable for publication, you may indicate that here to bypass the “Comments to the Author” section, enter your conflict of interest statement in the “Confidential to Editor” section, and submit your "Accept" recommendation.

Reviewer #1: All comments have been addressed

Reviewer #2: All comments have been addressed

2. Is the manuscript technically sound, and do the data support the conclusions?

Reviewer #1: No

Reviewer #2: Yes

3. Has the statistical analysis been performed appropriately and rigorously? 

Reviewer #1: No

Reviewer #2: Yes

4. Have the authors made all data underlying the findings in their manuscript fully available?

Reviewer #1: No

Reviewer #2: Yes

5. Is the manuscript presented in an intelligible fashion and written in standard English?

Reviewer #1: No

Reviewer #2: Yes

6. Review Comments to the Author

Reviewer #1: The response was not in satisfactory level.

Without proper justification of the point I can't accept the manuscript.

Reviewer #2: (No Response)

7. PLOS authors have the option to publish the peer review history of their article (what does this mean?). If published, this will include your full peer review and any attached files.

Reviewer #1: No

Reviewer #2: No

---

## [Editor Report · Acceptance letter]

8 Apr 2024

PONE-D-23-27697R1 

PLOS ONE

Dear Dr. Kwon, 

I'm pleased to inform you that your manuscript has been deemed suitable for publication in PLOS ONE. Congratulations! Your manuscript is now being handed over to our production team.

Kind regards, 

on behalf of

Dr. Charushila Yuvraj Kadam 

Academic Editor

PLOS ONE